# Can Healthy and Sustainable Dietary Patterns That Fit within Current Dutch Food Habits Be Identified?

**DOI:** 10.3390/nu13041176

**Published:** 2021-04-02

**Authors:** Samantha N. Heerschop, Sander Biesbroek, Elisabeth H. M. Temme, Marga C. Ocké

**Affiliations:** 1Division of Human Nutrition and Health, Wageningen University, P.O. Box 17, 6700 AA Wageningen, The Netherlands; samantha.heerschop@wur.nl (S.N.H.);sander.biesbroek@wur.nl (S.B.); liesbeth.temme@rivm.nl (E.H.M.T.); 2National Institute for Public Health and the Environment (RIVM), Postbox 1, Bilthoven, 3720 BA Utrecht, The Netherlands

**Keywords:** sustainable diets, dietary pattern, reduced rank regression, greenhouse gas emissions, blue water use, acceptability

## Abstract

This study investigated major healthy and sustainable dietary patterns in the Dutch population. Two 24-hour dietary recalls were collected in 2078 participants aged 19–79 years in the Dutch National Food Consumption Survey 2012–2016. Dietary patterns were identified using reduced rank regression. Predictor variables were food groups and response variables were Dutch Healthy Diet index 2015 (DHD15-index) score, greenhouse gas emissions (GHGE), and blue water use. Three patterns were discovered, including a “high fruit and vegetable dietary pattern”, a “low meat dietary pattern”, and a “high dairy, low fruit juices dietary pattern”. Diets in the highest quartile of these patterns had higher DHD15-index score than the average population. However, diets of the “high fruit and vegetable dietary pattern” were associated with higher dietary GHGE (14%) and blue water use (69.2%) compared to the average population. Diets of the “low meat dietary pattern” were associated with lower GHGE (19.6%) and higher blue water use (7.7%). Concluding, the “low meat dietary pattern” was the most healthy and sustainable dietary pattern in this population. The addition of blue water use as an environmental impact indicator shows the difficulty of finding existing dietary patterns that have low environmental impact in all determinants.

## 1. Introduction

Ongoing climate change emphasizes the need for new strategies to improve sustainability, as stated by the Paris Climate Agreement and the United Nations Sustainable Development Goals [1,2]. Greenhouse gas emissions (GHGE) are a major driver of global warming, and hence, reduction is key [3]. Food production and consumption contribute 20–30% to the total GHGE [4]. In addition, food production is a major determinant in biodiversity loss, land use, and fresh water use [5,6]. Therefore, shifting towards more sustainable diets is important and urgent.

According to the Food and Agriculture Organization (FAO) of the United Nations, sustainable diets have low environmental impacts, are culturally acceptable, and are nutritionally adequate, safe, and healthy for present life and future generations [7]. To identify healthy and sustainable diets, several studies modelled a priori dietary patterns based on nutritional guidelines and environmental impact data. For example, the EAT-Lancet diet was introduced as the healthy and sustainable reference diet that enables us to feed the world without exceeding planetary boundaries [8]. The EAT-Lancet dietary recommendations include high consumption of vegetables, fruit, whole grains, legumes, nuts, and unsaturated oils, moderate consumption of seafood and poultry, and minimal consumption of red meat, processed meat, added sugar, refined grains, and starchy vegetables. Large differences are observed when comparing the current “Western diet”, such as the Dutch diet, to this reference diet [9], and consequently, large changes need to be implemented to meet recommendations of the EAT-Lancet diet. As a first step towards such a healthy and sustainable diet, data-driven or a posteriori methods can be used to derive example dietary patterns present within a population [10]. These dietary patterns have proven acceptability by at least part of the population.

An often-used posteriori method to derive dietary patterns is the principal component analysis [11,12]. Another method, the reduced rank regression (RRR), is able to extract dietary patterns that are stronger associated with several particular effect measures (response variables), such as disease risk factors, by combining both a priori and posteriori techniques [13,14]. A previous study used the RRR to search for dietary patterns in a Dutch cohort, using as response variables the Dutch Healthy Diet 2015 index (DHD15-index) as a proxy for healthiness of the diet and GHGE as an environmental impact indicator [15]. In this study, the derived “plant-based diet” included high consumption of fruit, vegetables, and legumes and benefited health as well as the environment. However, the second derived “dairy-based diet”, including high consumption of dairy and nuts and seeds, was somewhat healthier but at the expense of higher GHGE.

Since the effects of sustainable diets should not exceed any planetary boundary, the focus should not be only on GHGE. Vellinga et al. (2019) showed that GHGE is highly correlated with acidification, eutrophication, and land use, but not with blue water use. Therefore, including blue water use in the analysis could potentially provide a more complete picture of environmental impact of a diet. Furthermore, to find healthy and environmentally sustainable dietary patterns that are achievable for the Dutch population, a representative study population with most recent dietary information is needed. Participants that completed the latest Dutch National Food Consumption Survey (DNFCS) (2012–2016) would be such a study population [16].

To gain insight in healthy and environmentally sustainable dietary patterns that are realistic and achievable for the Dutch population, this study investigated which dietary patterns were present in the study population that might be beneficial for health and the environment.

## 2. Materials and Methods

### 2.1. Study Population

The study population consisted of participants of the DNFCS [16]. This survey consisted of 4313 participants aged 1–79 years and was conducted between 2012–2016 by the National Institute for Public Health and the Environment (RIVM), the Netherlands. Participants were drawn from a representative consumer panel of the market research agency KANTAR TNS. Panel members participate in all types of studies. An age–gender random sampling strategy was applied. Furthermore, representativeness of region, address density, and education was taken into account. The response rate was 65%. The DNFCS was conducted according to the guidelines of the Helsinki Declaration. Because of non-invasive measurements in this survey, the Medical Ethical Committee of the University Medical Centre Utrecht, the Netherlands, concluded that the study did not need to be evaluated according to the “Medical research on human act” (WMO).

Children younger than 19 years were excluded from this study (*n* = 2235). The total population for analyses was aged 19–79 and consisted of 2078 participants.

### 2.2. Dietary Assessment

Trained dieticians collected food consumption data by two non-consecutive 24-hour dietary recalls. Standardized interviews were conducted using the GloboDiet (former EPIC-soft©) computer program, provided by the International Agency for Research on Cancer, Lyon, France [17]. Participants aged >70 years received an additional food recording booklet to be kept at the day before the call. To obtain consumption information representative for the whole year, the 24-hour dietary recalls were spread over seasons and both week and weekend days.

The energy of consumed products was derived from the Dutch Food Composition Database (NEVO—online version 2016/5.0) [18]. Food items from the DNFCS were grouped in 21 food groups adapted from the GloboDiet food group categorization. For the present study, consumptions were presented in g/2000 kcal. This standardization was to account for differences in energy intake by age and gender. In this way of standardization, food items that contain no calories, such as water, coffee, and tea, could still be taken into account in the analysis.

### 2.3. Assessment of Healthiness of Diets

The healthiness of a diet was scored using the DHD15-index [19]. This index distinguishes fifteen components, each representing one of the fifteen Dutch dietary guidelines of 2015 (Table A1) [20]. A score between 0, indicating no adherence, and 10, indicating complete adherence, was attributed to each component [15]. Since there was no information available on type of coffee consumed (filtered or not), this item was not taken into account. DHD15-index scores in this study could, therefore, potentially range between 0 and 140 points, where a score of 140 indicates maximal adherence to the guidelines.

### 2.4. Assessment of the Environmental Impact of Diets

GHGE (kg CO_2_ equivalents/2000 kcal) and blue water use (m^3^/2000 kcal) of food products consumed by the Dutch population were determined using Life Cycle Assessment (LCA) [21]. LCAs take into account the process of production, transportation, preparation, and waste or losses of a product at all stages of the life cycle. Blonk Consultants provided Life Cycle Inventories to estimate environmental impact of a product [22]. Vellinga et al. (2019) provided a more extensive description of definitions of GHGE and blue water use and of the usage of LCA in the DNFCS [23].

Table A2 shows median GHGE and blue water use per kg products within food groups. Calculations were based on consumption of participants, excluding non-consumers in each food group.

### 2.5. Lifestyle and Anthropometric Variables

A general questionnaire was used to derive information on covariates. For participants aged 19–70, height and weight were self-reported. From this information, body mass index (BMI) was calculated (kg/m^2^). For people aged >70, weight was measured and height was not reported; therefore, BMI could not be calculated. Educational level was classified as low (primary education, lower vocational education, advanced elementary education), moderate (intermediate vocational education, higher secondary education), or high (higher vocational education and university).

### 2.6. Statistical Analysis

Characteristics of the study population are presented as mean and standard deviation for continuous, normally distributed variables. For continuous, skewed data, median and interquartile range are shown. Categorical variables are shown as percentages.

RRR was used to extract dietary patterns that might benefit health and the environment. This method determines linear functions of predictors by maximizing the explained variation in various response variables [14]. To perform this analysis, the PROC PLS procedure in SAS was used. Predictor variables were food groups in gram per 2000 kcal. Response variables were DHD15-index score, daily dietary GHGE per 2000 kcal, and daily dietary blue water use per 2000 kcal. The number of dietary patterns derived is equal to the number of response variables. Participants received a pattern score for each pattern. Pattern scores were split in quartiles, and participants in quartile 4 (Q4) were the highest adherents to the pattern. If the highest adherents to a pattern showed lower DHD15-index than lowest adherents, pattern scores were multiplied by (−1) to obtain the healthier diet in Q4. Dietary patterns were labelled based on the two food groups that had the strongest association with the pattern. A factor loading of >|0.20| was considered important.

All statistical analyses were performed using SAS software (version 9.4, SAS Institute Inc., Cary, NC, USA). A two-sided *p* value of < 0.05 was considered statistically significant.

## 3. Results

### 3.1. Population Characteristics and Food Consumption

The adult population from the DNFCS consisted of 2078 participants of which 50.2% were male (Table 1). The median age was 51 years, and the median energy intake was 2064 (interquartile range (IQR): 1699–2552) kcal/day. The mean DHD15-index score was 59.4 (standard deviation (SD): 18.6) out of the potentially maximum score of 140. The median dietary GHGE per 2000 kcal was 4.7 (IQR: 4.02–5.62) kg CO_2_ equivalents, and the median dietary blue water use was 0.13 (IQR: 0.10–0.19) m^3^ per 2000 kcal.

### 3.2. Dietary Patterns Derived by RRR

Using RRR, three dietary patterns were derived. The first pattern, “high fruit and vegetable dietary pattern”, explained 37.5% of the variation in DHD15-index, dietary GHGE, and dietary blue water use and 8.9% of the variation in food consumption. The second pattern, “low meat dietary pattern”, explained 21.3% of the variation in the dependent variables and 4.7% in the predictor variables. The third pattern, “high dairy, low fruit juices dietary pattern”, explained 7.7 and 5.3% in the dependent and predictor variables, respectively.

#### 3.2.1. Healthiness and Sustainability of the Three Dietary Patterns

Diets in Q4 of the pattern scores of the “high fruit and vegetable dietary pattern” had a 27.4% higher DHD15-index score than the diets of the average Dutch population (75.6, SD: 15.3) versus 59.4 (SD: 18.6) (Table 2). These diets had a 14.0% higher dietary GHGE and 69.2% higher dietary blue water use than diets of the average Dutch population (5.36 (IQR: 4.54–6.33) kg CO_2_ eq/2000 kcal versus 4.70 (IQR: 4.02–5.62) CO_2_ eq/2000 kcal and 0.22 (IQR: 0.17–0.28) m^3^/2000 kcal versus 0.13 (IQR: 0.10–0.19) m^3^/2000 kcal, respectively). Diets in Q4 of the “low meat dietary pattern” had a 17.0% higher DHD15-index score, a 19.6% lower dietary GHGE, and a 7.7% higher dietary blue water use compared to diets of the average population (69.5 (SD: 17.8) versus 59.4 (SD: 18.6), 3.78 (IQR: 3.35–4.22) kg CO_2_ eq/2000 kcal versus 4.70 (IQR: 4.02–5.62) CO_2_ eq/2000 kcal and 0.14 (IQR: 0.09–0.20) m^3^/2000 kcal versus 0.13 (IQR: 0.10–0.19) m^3^/2000 kcal, respectively) (Table 2). With increasing adherence to the “high dairy, low fruit juices dietary pattern”, a 13.0% higher DHD15-index score, a 9.1% higher dietary GHGE, and a 7.7% lower dietary blue water use were observed (Table 2).

#### 3.2.2. Dietary Characterization of the Three Dietary Patterns

In the “high fruit and vegetable dietary pattern”, Q4 was characterized by high consumption of vegetables (factor loading (Fl): 0.51), fruit (Fl: 0.48), coffee and tea (Fl: 0.40), water (Fl: 0.22), and fruit and vegetable juices (Fl: 0.20), and low consumption of soft drinks (Fl: −0.22) (Figure 1). The high fruit and vegetable consumption and the low consumption of soft drinks within dietary pattern caused the high DHD15-index score compared to the average population. The relatively high consumption of dairy, fruit, vegetables, and coffee and tea in Q4 of the “high fruit and vegetable dietary pattern” caused the increased GHGE. Besides, the consumption of fruit, vegetables, and coffee and tea caused the high dietary blue water use in this pattern. See Table 1 for intake per food group in Q1 and Q4 per pattern and Table A2 for GHGE and blue water use per kg food group.

The “low meat dietary pattern” was defined as high consumption of sweets and snacks (Fl: 0.36) and nuts and seeds (Fl: 0.26), and the low consumption of red unprocessed meat (Fl: −0.65) and processed meat (Figure 1). The increased nuts and seeds consumption and the decreased red unprocessed and processed meat consumption caused the increased DHD15-index score compared to the average population. However, due to the increased consumption of sweets and snacks, a smaller increase in DHD15-index score is observed than in the “high fruit and vegetable dietary pattern”. The “low meat dietary pattern” had the lowest dietary GHGE, caused by relatively low processed and unprocessed red meat, dairy, vegetable, and fruit consumption (Table A2). Fruit, nuts and seeds, sweets and snacks, and coffee and tea consumption caused the slightly increased dietary blue water use compared to the average population (Table A2).

Q4 of the “high dairy, low fruit juices dietary pattern” was characterized as high consumption of dairy (Fl: 0.43) and potatoes and cereals (Fl: 0.23), and the low consumption of fruit and vegetable juices (Fl: −0.70), alcoholic beverages (Fl: −0.23), and nuts and seeds (Fl: −0.20) (Figure 1). The slightly increased dairy, fruit, and vegetable consumption and slightly decreased soft drinks and sweets and snacks consumption caused the increased DHD15-index score compared to the average population. The unprocessed red meat and dairy consumption in this diet was responsible for the slightly increased GHGE (Table A2), and the relatively low vegetable, fruit, and coffee and tea consumption caused the slightly decreased dietary blue water use compared to the average population and the other two dietary patterns (Table A2).

Fruit intake in Q4 of the “low meat dietary pattern” and the “high dairy, low fruit juices dietary pattern” differed only 0.8% (112 g versus 113 g, respectively), but blue water use of the food group “fruit” of diets in Q4 of the “low meat dietary pattern” was 36.4% higher than that of diets in Q4 of the “high dairy, low fruit juices dietary pattern” (0.015 m^3^ versus 0.011 m^3^, respectively). This reveals that types of fruit eaten differs between the patterns. The same is true for the food group coffee and tea in these two dietary patterns.

#### 3.2.3. Characteristics of Adherents of the Three Dietary Patterns

Adherents of the “high fruit and vegetable dietary pattern” (participants in Q4 of the pattern scores) were more likely to be older, to be female, and to be higher educated compared to the average population (59 vs. 51 years; 72.4 vs. 49.8% females; 37.0 vs. 33.1% higher educated, respectively) (Table 1). Besides, these adherents had lower energy intake and smoked less compared to the average population (1715 vs. 2064 kcal/day; 15.2 vs. 20.0% smokers). Adherents of the “low meat dietary pattern” tended to be higher educated (40.3 vs. 33.1% higher educated), to be younger (47 vs. 51 years), smoked less (14.3 vs. 20.0% smokers), had a slightly lower BMI (24.0 vs. 25.5 kg/m^2^), and had higher energy intake compared to the average population (2151 vs. 2064 kcal/day). Adherents of the “high dairy, low fruit juices dietary pattern” were older and had a lower energy intake compared to the average population (55 vs. 51 years; 1857 vs. 2064 kcal/day).

### 3.3. Differences in Pattern Scores per Level of Education

Table 3 shows pattern scores per level of education, stratified for age and gender and corrected for BMI. For the “high fruit and vegetable dietary pattern”, higher educated males and females of all ages had higher pattern scores than the lower educated groups. Besides, with increasing age, pattern scores for every educational level increased for both males and females. Females had higher pattern scores on this dietary pattern at every age and every level of education compared to males.

For the “low meat dietary pattern”, higher educated males and females younger than 40 years had higher patterns scores than the lower educated groups. Pattern scores for males and females of the ages 40–59 years and >59 years did not differ significantly per level of education.

Pattern scores for males and females of all age groups did not differ significantly per level of education for the “high dairy, low fruit juices dietary pattern”.

## 4. Discussion

In this study, three dietary patterns were derived from the Dutch National Food Consumption Survey 2012–2016 using RRR: a “high fruit and vegetable dietary pattern”, a “low meat dietary pattern”, and a “high dairy, low fruit juices dietary pattern”. The “low meat dietary pattern” was the most sustainable pattern with diets in Q4 having 19.6% lower GHGE and 7.7% higher blue water use compared to diets of the average population. Since these patterns are derived from food consumption information of the Dutch population, it may be assumed that these patterns are socially acceptable for at least part of the population. In any pattern, as observed in this study, a shift is possible towards healthier and environmentally sustainable diets. As yet, none of the patterns showed the optimal combination of increased DHD15-index score and decreased dietary GHGE and dietary blue water use.

Our results are generally in line with dietary patterns that were found in the EPIC-NL cohort using the RRR with DHD15-index and GHGE as dependent variables [15]. The “plant-based diet” derived from the cohort was healthier and had lower dietary GHGE compared to the average diet of that population. This pattern is a combination of the “high fruit and vegetable dietary pattern” and the “low meat dietary pattern”, with blue water as the additional separating environmental impact indicator in the current study. The “dairy-based diet” of the cohort was somewhat healthier and had higher dietary GHGE, which corresponds to our “high dairy, low fruit juices dietary pattern”. The same was observed in another study that searched for healthy and environmentally sustainable dietary patterns in five European countries, using a multiple factor analysis, focusing on GHGE and mean adequacy ratio, mean excess ratio, and solid energy density as proxies for nutritional value of the diet [24]. They found a diet that was healthier and more sustainable, in which significantly larger quantities of plant-based products and smaller quantities of meat, soft drinks, and alcoholic beverages were consumed. This is similar to a combination of our first two patterns with blue water use a separator. Another study that used a similar data-driven method and eight response variables, indicating health, environment, and affordability derived a dietary pattern, which included relatively low amounts of animal origin products, especially red meat, sweets, and fatty products, and substantial consumption of soy-based and whole products [25]. This pattern matches our “low meat dietary pattern” regarding low meat consumption. However, sweets and snacks consumption does not correspond, which might be caused by the different response variables used. Similar to our derived “low meat dietary pattern”, a review on the impact of dietary changes on the environment concluded that a reduction in animal-based foods would result in substantial reductions in diet-related GHGE, land use, and water use [26]. Another review states that reducing the amount of meat and changing the type of meat mainly affects the environmental improvement potential regarding GHGE and land use [27].

Comparing the dietary patterns derived from the Dutch population to the healthy and sustainable reference diet of the EAT-Lancet Commission, several similarities and differences are observed. [8]. Diets in the highest quartile of the “low meat dietary pattern” match the reference diet in the low red and processed meat consumption and the high consumption of nuts and seeds. However, these diets also show high consumption of sweets and snacks (adding in a limited way to environmental impact), which is in contrast with the EAT-Lancet reference diet. The high fruit and vegetable intake in diets in the highest quartile of the “high fruit and vegetable dietary pattern” do correspond with the reference EAT-Lancet diet, but the relatively high meat and dairy intake do not. The low intake of fruit and vegetable juices in the “high dairy, low fruit juices dietary pattern” is in line, but the high dairy consumption is not. Summarizing, the dietary patterns found in this study still show a distance from this EAT-Lancet reference diet. Even for the most beneficial “low meat dietary pattern”, there is much to gain regarding health and environmental impact aspects. A study about dietary changes that are needed to reach a healthy and environmentally sustainable diet in different European countries showed that GHGE could be theoretically decreased by 62–78%, while still being nutritionally adequate, but this is at a strong risk of compromising cultural acceptability of the diets [28]. Concluding, other European countries also still show a large distance from a healthy and environmentally sustainable dietary pattern. However, given the presence (21.3% of the variation in response variables explained) of the “low meat dietary pattern” in the Dutch population, aspects of this diet are achievable for at least part of the population. Despite Dutch diets being far from environmental sustainability yet, the “low meat dietary pattern” is a good starting point for developing realistic environmentally sustainable dietary patterns. To improve the health and sustainability of the “low meat dietary pattern”, guidelines may focus on decreasing sweets and snacks consumption and moderating nuts and seeds consumption, in order to reduce blue water use. As an alternative, consumption of, for example, legumes may be promoted, as they benefit both health and environment. Furthermore, a study that used data of the DNFCS 2007–2010 showed that choosing low GHGE foods from each food group within a healthy diet results in reductions in dietary GHGE that are comparable to reductions achieved in healthy diets without meat [29].If consumers besides low meat consumption choose low GHGE products from each food group, large reductions in GHGE of a healthy diet can be achieved, whether or not the same concept holds for blue water use can be a subject for future studies.

Diets in the highest quartile of the “high fruit and vegetable dietary pattern” have highest DHD15-index scores, but also show very high blue water use. It can be worthwhile to study which products cause the high blue water use in this pattern. For example, raspberries have a twelve times higher water use than apples. When using more specific food groups that are more homogeneous in dietary GHGE and blue water use, dietary patterns that optimize health and both environmental indicators might be revealed.

Observed differences when comparing pattern scores of dietary patterns per level of education were comparable to other studies. Biesbroek et al. (2018) found that the higher educated group had higher pattern scores on the “prudent dietary pattern”, which is comparable to our “high fruit and vegetable dietary pattern” [30]. This is in line with existing literature, which says that higher educated people have healthier diets according to the DHD15-index score and the consumption of energy, fat, fiber, fruit, vegetables, and energy-rich drinks, respectively, but the diets are less environmentally friendly [31,32]. When developing interventions or dietary guidelines, policy makers may take into account that higher educated people adhere more to a “high fruit and vegetable dietary pattern” and that young (<40 years) higher educated people adhere more to the “low meat dietary pattern”. E.g., in young (<40 years) higher educated consumers, the focus could be more on lower consumption of sweets and snacks of which consumption is high in the “low meat dietary pattern”. More specific interventions or dietary guidelines for subgroups might increase cultural acceptance and thereby compliance [33,34,35].

This study has several strengths. The first one is the usage of a second environmental impact indicator to provide a more complete picture of environmental impact of diets. The second strength is the hybrid approach of the RRR. This method allows us to find dietary patterns that are associated with the response variables of interest, if healthy and environmentally sustainable dietary patterns are present in the population, compared to the older and often-used principal component analysis [11]. Another strength is that food consumption information was based on two non-consecutive 24-hour dietary recalls. This method provides detailed food consumption information and is less subject to bias than food frequency questionnaires [36]. Lastly, the used food consumption information is from the most recent DNFCS, which is stratified for age, gender, region, address density, and level of education. This reflects the most current and representative diets of Dutch inhabitants.

Additionally, some limitations should be mentioned. Firstly, because of the addition of a second environmental impact indicator, it might be argued that environmental impact has a larger influence on the derived dietary patterns than health. However, intake in the highest quartile of all patterns that were found are healthier than the average diet. Secondly, despite a large percentage of the variation in the response variables was explained by the dietary patterns, namely 37.5, 21.3, and 7.7%, respectively, only 8.9, 4.7, and 5.3% of the variation in the predictor variables was explained. These percentages are comparable to other studies using RRR to identify dietary patterns [11,14,15]. Data envelopment analyses methods are currently under development and might be used in the future to derive dietary patterns, since this method can maybe explain a larger percentage of the variation in predictor variables [37,38]. A third limitation is the fact that not all environmental impact indicators are included in the study. An important missing indicator is biodiversity loss [39]. Because of the lack of data, we were not able to include this in our analysis. The addition of data on biodiversity loss will improve completeness of the impact of the diet on the environment. As GHGE is highly correlated (rho > 0.7) with acidification, fresh water eutrophication, marine eutrophication, and land use, a wide range of environmental impact indicators is indirectly taken into account in this study [23]. Another limitation is the large amount of non-consumers and the possible misreporting of the self-reported recalls [36,40]. However, assuming that misreporting is independent of specific food groups, due to the standardization for energy intake, part of the misreporting is corrected for [41]. Besides, to obtain a perfect representation of the Dutch population, a weighing factor was desirable to add to the RRR model, but SAS did not provide an option to add this weighing factor. However, with only small deviances from the real population, our study population was a good representation for the Dutch population [16]. Furthermore, despite the fact that the most recent food consumption survey data were used in this study, eating habits might already have changed between 2016 and 2021. The most recent trends in dietary pattern are not taken into account as food consumption surveys are time-bounded. A study on the acceptance of alternative protein sources for meat concludes that Dutch consumers have a higher acceptance towards all alternative proteins in 2019 compared to 2015. However, self-reported consumption of alternative proteins shows no differences across years [42]. Therefore, using data that are representative of the Dutch population at 2012–2016 is still insightful. Another limitation is that food groups and the response variables dietary GHGE and blue water use were standardized for energy intake, but not the DHD15-index. The latter measure is based on absolute consumption, and standardization would violate the true score. Since two out of three response variables were standardized, results might be slightly distorted. However, using only unstandardized variables would result in dietary patterns based on variations in diet quantity, and not in diet quality. The last limitation of this study is the unresolved uncertainty in LCA data. Unresolved problems of LCAs are, for example, spatial variation and local environmental uniqueness [43]. Primary LCA data were available for 242 food products and cover 71% of the quantity consumed in the DNFCS. Remaining food products (29%) are based on extrapolated data. However, due to the extrapolations, our LCA data are complete [23]. Besides, LCAs are the best estimates available for environmental impact of foods, though they always include uncertainties.

## 5. Conclusions

To the best of our knowledge, this is the first study that provides insight in existing dietary patterns in a representative study population for the Dutch society, using the reduced rank regression analysis. Three socially acceptable dietary patterns were extracted: the “high fruit and vegetable dietary pattern”, the “low meat dietary pattern”, and the “high dairy, low fruit juices dietary pattern”. In any of these patterns, a shift is possible towards healthier and environmentally sustainable diets. However, none of the patterns showed the optimal combination of increased DHD15-index score and decreased dietary GHGE and dietary blue water use. The “low meat dietary pattern” was the healthiest and most environmentally sustainable pattern with diets in the highest quartile having 17.0% higher DHD15-index score, 19.6% lower GHGE, and 7.7% higher blue water use. The addition of blue water use as an environmental impact indicator in this study shows the difficulty of finding existing dietary patterns that have low environmental impact in all determinants. Future research might focus on the role of foods or food groups in dietary patterns where health and/or different environmental impact indicators do align to optimize dietary patterns that are socially acceptable, healthy, and sustainable.

## Figures and Tables

**Figure 1 nutrients-13-01176-f001:**
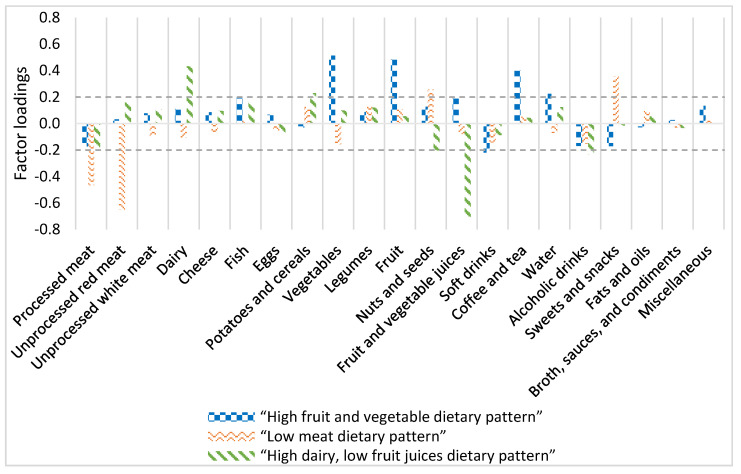
Factor loadings of the food groups on three dietary patterns derived by reduced rank regression explaining the variation in the Dutch Healthy Diet 2015 index scores, dietary greenhouse gas emission, and dietary blue water use. Factor loadings >|0.20| were considered important contributors to a dietary pattern. Note: of the “low meat dietary pattern”, pattern scores of participants and factor loadings were multiplied by (−1) to obtain the healthier and more sustainable diet in quartile 4.

**Table 1 nutrients-13-01176-t001:** Characteristics and consumption (g/2000 kcal) per food group of the total adult population (*n* = 2078) and diets in quartile one and four of the “high fruit and vegetable dietary pattern”, the “low meat dietary pattern”, and the “high dairy, low fruit juices dietary pattern”, derived by the reduced rank regression.

	Total Population from DNFCS	“High Fruit and Vegetable Dietary Pattern”		“Low Meat Dietary Pattern” ^1^		“High Dairy, Low Fruit Juices Dietary Pattern”	
		Quartile 1	Quartile 4	Quartile 1	Quartile 4	Quartile 1	Quartile 4
Age (years) (median, IQR)	51 (31–70)	41 (27–56)	59 (38–72)	56 (36–71)	47 (30–68)	49 (30–68)	55 (31–72)
Males (*n* (%))	1043 (50.2)	336 (64.7)	143 (27.6)	263 (50.7)	250 (48.2)	278 (53.6)	240 (46.2)
Body mass index (kg/m^2^) ^2^ (median, IQR)	25.5 (22.7–29.0)	25.2 (22.2–29.0)	25.6 (22.9–29.4)	27.2 (24.2–30.6)	24.0 (21.7–27.1)	25.3 (22.7–28.4)	25.9 (23.3–29.5)
Smokers (*n* (%))	413 (20.0)	136 (26.6)	79 (15.2)	121 (23.4)	74 (14.3)	132 (25.6)	93 (18.0)
Energy intake (kcal/day) (median, IQR)	2064 (1699–2552)	2459 (1968–2956)	1715 (1421–2020)	1922 (1562–2421)	2151 (1772–2660)	2170 (1809–2679)	1857 (1547–2300)
Education (*n* (%)) ^3^							
Low	602 (29.0)	145 (27.9)	160 (30.8)	171 (33.0)	126 (24.3)	133 (25.6)	172 (33.1)
Moderate	789 (38.0)	234 (45.1)	166 (32.0)	207 (39.9)	184 (35.5)	210 (40.5)	217 (41.8)
High	687 (33.1)	140 (27.0)	193 (37.2)	141 (27.2)	209 (40.3)	176 (33.9)	130 (25.1)
Dietary consumption (gram/2000 kcal)							
Animal-based products							
Meat							
Processed meat	35.5 (11.7–67.5)	49.4 (24.6–84.4)	20.8 (0–55.4)	62.43 (26.0–103.6)	12.9 (0–31.3)	44.9 (16.3–77.4)	21.8 (7.6–46.1)
Red unprocessed meat	20.3 (0–53.0)	18.3 (0–46.1)	18.0 (0–57.2)	71.3 (29.9–104.7)	0 (0–16.4)	9.6 (0–38.1)	36.1 (0–75.8)
White unprocessed meat	0 (0–23.4)	0 (0–17.4)	0 (0–33.6)	0 (0–24.3)	0 (0–16.7)	0 (0–16.1)	0 (0–37.7)
Dairy	255.0 (120.0–421.8)	191.1 (65.8–336.4)	303.4 (145.0–485.5)	275.6 (122.2–458.2)	221.8 (95.7–365.2)	148.7 (46.0–270.6)	453.0 (292.3–613.4)
Cheese	28.2 (12.8–48.2)	22.3 (7.5–38.7)	32.8 (17.1–55.0)	28.4 (11.8–49.9)	23.3 (10.8–42.1)	25.7 (8.5–44.6)	32.8 (16.6–56.9)
Fish	0 (0–14.9)	0 (0–0)	0 (0–52.0)	0 (0–0)	0 (0–23.6)	0 (0–0)	0 (0–54.0)
Eggs	0 (0–22.9)	0 (0–17.3)	0 (0–26.2)	0 (0–24.4)	0 (0–21.1)	0 (0–24.1)	0 (0–16.6)
Plant-based foods							
Potatoes and cereals ^4^	256.4 ± 85.0	247.7 ± 78.3	249.7 ± 92.8	242.9 ± 89.1	263.1 ± 84.6	227.8 ± 80.5	293.7 ± 89.2
Vegetables	125.6 (73.8–204.1)	65.6 (34.6–100.0)	237.8 (164.5–329.5)	152.7 (90.0–234.7)	109.3 (57.2–189.4)	112.2 (56.8–190.1)	150.5 (94.3–236.0)
Legumes	0 (0–0)	0 (0–0)	0 (0–0)	0 (0–0)	0 (0–0)	0 (0–0)	0 (0–0)
Fruit	95.2 (13.5–193.7)	13.5 (0–65.2)	223.9 (136.59–346.8)	76.7 (0–171.1)	112.4 (37.1–231.2)	83.0 (0–179.9)	113.1 (23.5–214.4)
Nuts and seeds	0 (0–13.1)	0 (0–6.1)	0 (0–20.9)	0 (0–1.4)	7.9 (0–26.1)	0 (0–24.9)	0 (0–0)
Beverages							
Non-alcoholic beverages							
Fruit and vegetables juice	0 (0–81.7)	0 (0–153.0)	0 (0–0)	0 (0–96.9)	0 (0–68.7)	125.6 (3.0–223.5)	0 (0–0)
Soft drinks	121.3 (0–360.3)	324.3 (87.7–653.3)	0 (0–171.4)	127.7 (0–406.6)	74.4 (0–254.3)	152.8 (0–390.3)	84.5 (0–276.9)
Coffee and tea	735.9 (452.2–1146.8)	430.9 (218.5–681.8)	1216.7 (777.6–1742.6)	715.1 (442.9–1141.9)	778.1 (479.9–1223.8)	706.4 (390.8–1106.8)	779.1 (493.4–1189.3)
Water	464.1 (149.4–956.1)	291.7 (65.1–704.6)	730.2 (318.0–1269.3)	504.1 (156.4–1151.4)	436.3 (144.1–918.7)	367.9 (127.5–816.1)	560.8 (201.1–1265.8)
Alcoholic beverages	0 (0–211.2)	62.8 (0–301.4)	0 (0–111.5)	10.8 (0–261.7)	0 (0–131.6)	94.4 (0–351.4)	0 (0–68.5)
Miscellaneous							
Sweets and snacks	71.8 (40.6–109.7)	85.0 (47.7–129.0)	56.8 (25.4–87.1)	48.5 (24.4–80.3)	99.5 (61.1–142.4)	68.8 (39.8–107.0)	64.0 (33.9–103.7)
Fat and oils	20.3 (12.6–28.8)	12.0 (12.5–29.0)	19.0 (11.1–28.3)	19.3 (11.6–27.3)	22.0 (13.7–31.7)	18.9 (11.6–27.2)	21.0 (12.7–29.2)
Broth, sauces, and condiments	50.0 (19.2–118.8)	52.9 (21.9–115.0)	46.2 (14.6–126.9)	47.0 (17.6–113.1)	46.1 (15.5–115.7)	48.7 (20.7–116.8)	39.2 (14.4–111.4)
Other	0 (0–0.2)	0 (0–0)	0 (0–1.6)	0 (0–0.7)	0 (0–0.1)	0 (0–0.2)	0 (0–0.1)

Median and interquartile ranges are shown for continuous variables. Categorical variables are presented as percentages. DNFCS: Dutch National Food Consumption Survey 2012–2016. IQR: interquartile range. ^1^ Pattern scores of participants are multiplied by (−1) to obtain the healthier and more sustainable diet in quartile 4. ^2^ From participants aged >70 years, no information on height was obtained. Therefore, body mass index is missing for these participants. Number of missing data on body mass index: total population of the DNFCS: 516. “High fruit and vegetable dietary pattern” quartile 1: 60, quartile 4: 172, “low meat dietary pattern” quartile 1: 147, quartile 4: 114, “high dairy, low fruit juices dietary pattern” quartile 1: 118, quartile 4: 150. ^3^ Level of education: low (primary education, lower vocational education, advanced elementary education), moderate (intermediate vocational education, higher secondary education), high (higher vocational education and university). ^4^ Presented as mean ± standard deviation.

**Table 2 nutrients-13-01176-t002:** Mean ± standard deviation Dutch Healthy diet index 2015 (DHD15-index) score and median (interquartile range) dietary greenhouse gas (GHG) emissions and dietary blue water use of participants in quartile 1 and 4 of the three derived dietary patterns compared to the total adult population of the Dutch National Food Consumption Survey (DNFCS) 2012–2016.

	Total Population of the DNFCS	“High Fruit and Vegetable Dietary Pattern”		“Low Meat Dietary Pattern” ^a^		“High Dairy, Low Fruit Juices Dietary Pattern”	
		Quartile 1	Quartile 4	Quartile 1	Quartile 4	Quartile 1	Quartile 4
DHD15-index score ^b^	59.4 ± 18.6	42.1 ± 13.1	75.6 ± 15.3	51.5 ± 17.3	69.5 ± 17.8	52.2 ± 19.0	67.1 ± 16.9
GHGE (kg CO_2_ equivalents/2000 kcal)	4.70 (4.02–5.62)	4.26 (3.70–4.98)	5.36 (4.54–6.33)	5.98 (5.20–6.96)	3.78 (3.35–4.22)	4.52 (3.78–5.49)	5.13 (4.42–6.05)
Blue water use (m^3^/2000 kcal)	0.13 (0.10–0.19)	0.09 (0.07–0.11)	0.22 (0.17–0.28)	0.13 (0.10–0.19)	0.14 (0.09–0.20)	0.18 (0.12–0.24)	0.12 (0.09–0.17)

^a^ Pattern scores of participants are multiplied by (−1) to obtain the healthier and more sustainable diet in quartile 4. ^b^ DHD15-index score: score out of 140 points.

**Table 3 nutrients-13-01176-t003:** Mean pattern scores per level of education, stratified for age and gender. Participants from the Dutch National Food Consumption Survey 2012–2016.

			Level of Education		
Dietary Pattern	Gender	Age	Low	Middle	High
“High fruit and vegetable dietary pattern” (pattern scores range between −2.913 and 7.789)	Male	<40	−1.033 ^a^ (*N* = 38)	−0.748 ^a^ (*N* = 141)	−0.436 ^b^ (*N* = 159)
		40–59	−0.508 (*N* = 53)	−0.479 (*N* = 134)	−0.250 (*N* = 104)
		>59	−0.323 ^ab^ (*N* = 47)	−0.412 ^a^ (*N* = 52)	0.031 ^b^ (*N* = 56)
		>70 *	*N* = 104	*N* = 79	*N* = 76
	Female	<40	−0.350 ^a^ (*N* = 39)	−0.216 ^a^ (*N* = 154)	0.434 ^b^ (*N* = 160)
		40–59	0.120 ^a^ (*N* = 81)	0.118 ^a^ (*N* = 136)	0.892 ^b^ (*N* = 65)
		>59	0.371 ^a^ (*N* = 240)	0.604 ^a^ (*N* = 93)	1.631 ^b^ (*N* = 67)
		>70 *	*N* = 167	*N* = 50	*N* = 40
“Low meat dietary pattern” (pattern scores range between −5.689 and 2.633) **	Male	<40	−0.039 ^ab^	−0.063 ^a^	0.247 ^b^
		40–59	−0.043	−0.146	0.076
		>59	0.003	−0.103	−0.081
	Female	<40	−0.210 ^a^	0.073 ^ab^	0.229 ^b^
		40–59	−0.048	0.011	0.042
		>59	−0.022	−0.045	0.303
“High dairy, low fruit juices dietary pattern” (pattern scores range between −5.638 and 2.834) **	Male	<40	−0.170	−0.014	−0.125
		40–59	−0.193	−0.083	−0.118
		>59	0.069	−0.174	−0.280
	Female	<40	−0.064	−0.130	0.013
		40–59	0.194	0.044	−0.032
		>59	0.262	0.094	0.085

Analysis of covariance: different superscript letters showing significant differences between pattern scores within a row (stratum). Pattern scores are adjusted for body mass index. * From participants aged > 70 years, no information on height was obtained. Therefore, body mass index is missing for these participants, and they are left out of this analysis. ** Number of participants per age, gender, and level of education is similar to the “high fruit and vegetable dietary pattern”.

## Data Availability

Data of the Dutch National Food Consumption Survey 2012–2016 can be requested for at https://www.rivm.nl/en/dutch-national-food-consumption-survey/data-on-request(accessed on 16 October 2019). Primary environmental data of 250 food products can be found at https://www.rivm.nl/voedsel-en-voeding/duurzaam-voedsel/database-milieubelasting-voedingsmiddelen (accessed on 16 October 2019).

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
