# Peer review of "Can Healthy and Sustainable Dietary Patterns That Fit within Current Dutch Food Habits Be Identified?"

_nutrients, 2021, doi:10.3390/nu13041176_

Round 1
Reviewer 1 Report
Major comments
The aim of this study is of great importance and the study is well designed. However, I have some major and minor comments.
- Title. I believe the title is misleading based upon the conclusions drawn by the authors. Sustainable dietary patterns were not identified - rather the results shows that there are no sustainable dietary patterns with the current eating habits in the Netherlands today. By choosing the most sustainable pattern – “low meat dietary pattern” - only minor improvements on e.g. blue water use (7.7%) and health (17%) was achieved.
- I recommend the authors to define sustainability - as written in the manuscript it appears they are only considering environmental aspects in their definition in contrast to e.g. FAO/WHO.
- Line 378 and on. Limitations. In my opinion the major limitation is that only two environmental aspects are included. I fully understand that this is because of lack of data but please mention it in the manuscript. In particular, the lack of biodiversity loss should be mentioned and the consequences of not adding such an environmentally important parameter discussed.
I also lack a discussion regarding the changes in eating habits since 2012-2016 when the Dutch food consumption survey was carried out. E.g. the sale of vegetarian products has risen significantly in many European countries and many Dutch food producers are active in this food segment and I would be very suprised if it would not have influenced the food patterns in the Netherlands during the last 5-9 years.
- Line 343-346. “..low meat dietary pattern” is a good starting point for developing realistic sustainable dietary patterns." I agree with the conculsion but it would improve the manuscript if it was nuanced stating that there is still a long way to go for a sustainable diets and elaborate on what other change that are needed for this pattern to be sustainable.
Minor comments
Line 52. Reduced rank regression is not a relatively new method, it was first published nearly 50 years ago. Even the application on dietary patterns is not relatively new, it was introduced 15 years ago and many publications using RRR to identify dietary patterns has been published thereafter. Please correct this.
Line 75. Study population, please add information on response rate.
Line 155. Reference error
Table 1.
Table heading. Please add the information that also data for quartile 1 is added in the table.
Body mass index – information on median and IQR is missing.
Food group “potatoe and cereals” – what is the rationale for combining potatoe and cereals? Considering the health benefits as well as environmental sustainability it would have been very interesting to present this food group divided into: potatoe, cereals excluding wholegrains and wholegrains.
For legumes it appears to be nearly no consumers – is this correct? Could it be due to the dietary assessment method? Or is it because the survey is 5-9 years old or do you believe the actual intake in the Dutch population is so low for legumes?
Line 389. Please separate misreportingof into two words.
Author Response
Dear editor,
Please find enclosed our revised manuscript entitled “Can healthy and sustainable dietary patterns that fit within current Dutch food habits be identified?” (nutrients-1165052). We thank you to reconsider this revised manuscript as an original contribution for publication in Nutrients.
The reviewer’s comments have all been addressed in the attached document 'Rebuttal reviewer 1', and included in the paper.

Reviewer 2 Report
Line 155 please add a reference.
"aged 1-79 years" is it correct?
Table 1 males - please provide a number next to the percentage, the same for educational level and smokers
Author Response
Dear editor,
Please find enclosed our revised manuscript entitled “Can healthy and sustainable dietary patterns that fit within current Dutch food habits be identified?” (nutrients-1165052). We thank you to reconsider this revised manuscript as an original contribution for publication in Nutrients.
The reviewer’s comments have all been addressed below, and included in the paper.
Point 1: Line 155 please add a reference.
Response 1: Done (line 156).
Point 2: "aged 1-79 years" is it correct?
Response 2: The complete dataset existed of 4313 subjects between the age of 1-79. However, in our study we only took the adult population into account (aged 19-79). See lines 90-91.
Point 3: Table 1 males - please provide a number next to the percentage, the same for educational level and smokers
Response 3: done (table 1).